# Inclusionary Leadership-Perspectives, Experiences and Perceptions of Principals Leading Autism Classes in Irish Primary Schools

Linda Dennehy *, Kevin Cahill and Joseph A. Moynihan

School of Education, University College Cork, T12 K8AF Cork, Ireland; kevin.cahill@ucc.ie (K.C.); joseph.moynihan@ucc.ie (J.A.M.)
* Correspondence: 119226432@umail.ucc.ie

**Abstract:** This study explores the experiences, practices and perceptions of primary school principals currently leading autism classes in Ireland. Autism classes in mainstream primary schools are becoming increasingly common in the Irish education system. The prevalence of autism classes highlights the importance of their role in enabling autistic children to attend mainstream schools. It reflects the increasing number of autistic pupils who require these specialised placements. Primary schools serve all children. It is essential that autistic children are supported in the best way possible so they can reach their full potential. The principal has a pivotal role in all aspects of his or her school, including leading the autism classes. Given the centrality of their role, it is imperative that the principal is supported by the best practices and theory available. This study sought to give the principals time to reflect on their inclusive leadership and decipher what it meant for them in their lived experience and context. Theories of leadership through a socio-cultural lens frame the overall study. A qualitative research design was adopted using semi-structured interviews with 15 primary school principals. Analysis of the data was conducted using a reflective thematic analysis approach. Findings of the research reveal that there are particular leadership styles that align with an inclusive leadership approach. These styles are distributed leadership, transformational leadership and instructional leadership. A positive disposition towards inclusion is an important factor in the principal's perceptions of their leadership. The idea of inclusionary leadership is borne out of the study. This term indicates that leaders striving for inclusion in their schools do not view it as a destination to be reached but rather a long-term journey they travel. This research is a pathway for further study in the field. It has implications for pupils, principals, school communities and policy makers regarding the value of the work of inclusionary leaders. All participants referred to in this paper have been given two letter pseudonyms to protect their identity.

**Keywords:** inclusion; principal; leadership; autism; autism class; primary schools; inclusionary leadership

## 1. Introduction

The educational landscape in Ireland is changing at a rapid rate with an influx of pupils from different faiths, cultures and an increase of children with additional needs attending mainstream schools. In particular, special education has experienced a number of changes and reforms in recent times [1]. With these changes comes the need for school principals to be proactive and innovative in developing their own leadership skills to respond to the rapidly changing landscape. While reviewing the literature, it becomes evident that there is no unanimous agreement on a definitive definition in the quest for the ideal inclusive leadership style or competency. Nevertheless, the United Nations Convention on the Rights of Persons with Disabilities (UNCPRD) emphasises that at the heart of inclusive education lies "the committed leadership of educational institutions", which is crucial for introducing and ingraining the culture, policies, and practices necessary to achieve inclusive education at all levels [2] (p. 4). The relationship between student

outcomes or attainment is clearly linked to leadership [3]. Given that school principals have an impact on student outcomes [4] it stands to reason that gaining an insight into the experiences, perspectives and perceptions of the principal will support the inclusion of autistic pupils [5]. This paper examines the perspectives of Irish primary school principals, when leading autism classes and trying to affect positive change [6].

### 1.1. Leadership for Inclusion in an Irish Context

Murphy [7] contends that more research is needed within the realm of inclusive leadership in Ireland in order to support its increasing diversity and to sustain leadership in the long-term. Ireland's approach to special educational needs in the past was a system whereby general education and special education developed simultaneously but parallel to one another [1,8]. The importance of the principal in the enactment of inclusive pedagogy in Irish schools is clear [9]. Research has found that Irish principals "demonstrated a clear commitment to inclusive education . . . in their schools" [10] (p. 1002). Principals play a pivotal role in ensuring resources are used prudently to support all students [11]. Shevlin and Banks [8] reviewed the current model of special education provision. They suggest that a different model may be in place in the future. This highlights the importance for Irish principals in the development of a framework for inclusive leadership that can adapt to any changes necessary to ensure pupils are supported in the best way possible.

### 1.2. Language of Inclusive Leadership

A discrepancy that arises from the literature is the phraseology of inclusive leadership versus leading for inclusion. While the interchange of the wording appears simplistic and unremarkable, the phrases denote different ideals. Are principals leading in an inclusive manner or is their leadership leading to inclusion? It could be argued that the result is the same, although the process differs. This differential may be evident within the skills used by the principal in their leadership. The language used to define the skills employed by a leader to carry out their duties varies within the literature. Cobb [12] likens the work of a principal to that of a Shakespearean actor who performs as required for a given audience. The skills needed by principals and leaders can be framed within "domains" [12,13]. This term is indicative of ownership or a territorial expression suggesting that leadership is an absolute. This idea of singularity negates the idea of shared leadership as central to the running of any school [13]. Óskarsdóttir, Donnelly [14] use the term "models of leadership" (p. 527) in reference to inclusive leadership. This terminology denotes the idea of modelling a standard for leadership that can be imitated by others. The idea proposed emerges from a review of a variety of different leadership models and focuses on those best suited to support inclusion. Competencies is another term visible within the literature to describe the skills used by a leader to enact their leadership. Competencies can be defined as a person's ability to do something well or effectively [15]. While the term is certainly apt for what we hope leaders strive for, what is less clear is how inclusion is achieved. Inclusive schools are described as needing "input variables" and "essential processes" to operate [16] (p. 1351). When looking towards inclusive leadership we need to explore the competencies and leadership styles that best promote this type of leadership.

### 1.3. Roles That Principals Play When Leading for Inclusion

When identifying the role of the leader within the research we must first look to what is specifically required of a leader to undertake within the area of inclusion. Fitzgerald [17] discusses the role of the principal within the framework of the Inclusion of Students with Special Educational Needs Guidelines [18]. Long [19] discusses leadership as a key factor in the provision of an inclusive education. Much of the literature speaks to leadership as being central to inclusive education [20]. Cobb [12] states that the work which principals do in relation to special education is "crucial" (p. 221). Within the literature, the roles and responsibilities that the principal must undertake are vast and it can be difficult to countenance a specific list. Five roles that emerged from the meta-analysis of inclusive

leadership literature for inclusive program delivery within these domains are those of: visionary, advocate, innovator, interpreter and organizer [12] (p. 221). Fitzgerald [20] outlines key positions that a leader must assume within the school: "arbiter, rescue[r], auditor, collaborator and expert" [20] (p. 454). Cobb [12] suggests that within the area of staff collaboration, the principal must take on the roles of "visionary, partner, coach, conflict resolver and organizer." [12] (p. 223). Under the domain of parental engagement, Cobb [12] identified three principal roles: partner, interpreter and organizer (p. 227). Kinsella [16] contends that there are five core processes of inclusion: communication, consultation, collaboration, co-ordination and collaborative enquiry. Given the wide-ranging tasks associated with leading for inclusion and in an effort to identify the most common roles within the research, the results are tabulated in the Table 1 from a number of recent research studies.

**Table 1.** Roles principals play according to the literature.

| | Inclusive Programme Delivery. Cobb (2015) [12] | Staff Collaboration. Cobb (2015) [12] | Parental Engagement. Cobb (2015) [12] | SENCO[1] Role in Post-Primary Schools in Ireland. Fitzgerald (2017) [20] | Organising Inclusive Schools. Kinsella (2020) [16] | SENCO & Principal's Experiences. Fitzgerald and Radford (2020) [10] |
|---|---|---|---|---|---|---|
| Visionary | ✓ | ✓ | | ✓ | | |
| Partner/ Collaborator/ mentor | | ✓ | | ✓ | ✓ | ✓ |
| Coach | | ✓ | | | | ✓ |
| Conflict re-solver/Arbiter | | ✓ | ✓ | ✓ | | |
| Advocate | ✓ | | | | | |
| Interpreter | ✓ | | ✓ | | | |
| Organiser/ coordinator | ✓ | ✓ | ✓ | ✓ | ✓ | |
| Consulter | | | | | ✓ | |
| Communicator | | | | | ✓ | |
| Rescuer | | | | ✓ | | |
| Expert | | | | ✓ | | ✓ |

The most common role identified across a variety of literature was principal as organizer/coordinator. Kinsella [16] contends that the organizational psychology paradigm is key to organizing inclusive schools. Within this paradigm the principal can be identified as the organizer of the program used for inclusive delivery. They organize how the school develops special education delivery within their context. The principal organizes the resources, timetabling and programs of staff professional development. They organize and carry out staff recruitment and foster staff teamwork through reflection and collaboration [10]. They assist parents by organizing supports and responding to the needs of the given cohort of children at any one time [12]. Fitzgerald [20] agrees that principals are responsible for organizing staff development and improvements in pedagogy. The identification of the organizer as a key role of the principal in inclusive education must be considered when developing a framework for inclusive leadership within the sector.

Another role that appears commonly across the literature is that of principal as partner/collaborator/mentor. This describes the principal as an intermediary for parents and

the services or government departments, to decipher and understand how to access the services that their child may need. Having examined the most common roles for inclusive leadership, it is necessary to review what current models of inclusive leadership exist in current practice.

### 1.4. Defining a Model of Inclusive School Leadership

Models of leadership are used to describe leadership practices rather than specific roles. They attempt to conceptualize or define leadership through particular characteristics or their utilization within leadership. Three key leadership styles are purported to be the most appropriate for the inclusive school namely transformational leadership, distributed leadership and instructional leadership [14].

### 1.5. Transformational Leadership

Transformational leadership as a style emerges frequently within the leadership for inclusion research [3,14,21]. This type of leadership is traditionally associated with the ability to affect change and innovation by impacting people and cultures within schools [22,23]. This type of leadership may be important for a school leader who is trying to influence a change of culture within a school such as developing an inclusive culture. Woodcock and Hardy [24] describe inclusion as a *re-culturing* process which looks to change the values of the school. This cultural change affects how teachers see their work and their pupils [25]. A leader must firstly identify their own vision and apply this to a vision for the school and its community. This is akin to the idea that leadership is a practice in influencing others to follow the path to a shared goal [26].

A principal who wants to change and develop a school culture needs a strong background of professional development and experience, in order to justify what many may not believe in. It is this bravery that will make the difference and transform a school over time to best serve its pupils [27].

### 1.6. Distributed Leadership

Distributed leadership has been considered the "gold standard" of school leadership and has grown in popularity in recent years [28]. While there is a lack of consensus as to a definition of distributed leadership [29], in this instance it may be understood as an idea for sharing workload and responsibilities [30]. It is claimed in research that distributed leadership is the "default leadership response implemented by schools to manage increased pressure", for example during the COVID 19 pandemic crisis [29] (p. 388). Spillane [31] cautions that distributed leadership is not just the sum of its characteristics, but also the result of the interactions between people and their situations. Jones and Harris [32] further this point by connecting the development of social capital and distributed leadership. This leadership type proffers that every member of the school is a leader in their own right Óskarsdóttir, Donnelly [14]. This can have a positive impact on student achievement and teacher job satisfaction [33]. It concentrates on expertise and not position reflecting schools as professional organizations where the knowledge base is wide [34]. The shared responsibility within a school increases the prospect of investment from staff and may increase the likelihood that an inclusive culture could be formed within a school. Hickey, Flaherty [29] found that there is ambiguity around the understanding of the sharing of leadership amongst staff. Distributed leadership can be misjudged or oversimplified as dividing out the work or delegation of simpler tasks where leaders may be perceived by staff as not carrying out their role to the full extent [35].

The lack of empirical evidence of the impact and effectiveness on educational outcomes from distributed leadership may be considered as one of this model's weaknesses [31]. However, if this model is encouraged by the principal, it would allow for leadership development within staff [33]. They will become leaders in their own professional journey and a culture of collaborative professional learning and development can ensue [14,34].

### 1.7. Instructional Leadership

This model of school leadership emphasizes the importance of curriculum through goal setting and evaluation of teaching and learning [14,36]. Much of the research shows that the most effective way to improve learner experience is to improve teachers' pedagogy [37]. Instructional leaders do this by putting the structures in place to allow teachers to improve their practice through collaboration and innovation [23,38]. A challenge for many principals is that they want to be instructional leaders but are overwhelmed by their administrative responsibilities [39]. There is a danger that they may be viewed by teachers as imposing ideas unless teachers are a key part of the process. Collaboration and the leaders' role in facilitating it, is seen as a key attribute of the instructional leader [3]. Positive collaboration can lead to improvements in teaching such as the use of the Individualized Educational Plan (IEP) with pupils and therefore impacts on learner experience [3]. As an instructional leader the principal needs a strong background in pedagogy themselves, so that they may have a thorough understanding of the curriculum. They have high expectations for all students including those with additional needs [40].

### 1.8. Combining Models of Leadership

While leaders themselves would identify that they may have a dominant leadership style, this style may vary depending on the situation [41]. Variance in the employment of different styles can be seen in the literature. While various styles were identified, no particular style was highlighted as being more inclusive than any other. What appears to be more commonly evident, is the idea of combining styles of leadership in order to better implement inclusive education [3]. Óskarsdóttir, Donnelly [14] has considered an Inclusive leadership style which combines the previously mentioned three styles and highlights how these co-exist with three necessary practices for leadership: building a vision, human development and organizational development. The idea of organizational development as a key practice reflects the previous literature that identified it as an integral role and responsibility of the principal for inclusive leadership. Setting direction speaks to the idea of culture and developing the inclusive culture of a school for its community. Human development speaks to developing the human resource capital within the school through professional development and organizational systems. It is clear to see from this vision of inclusive leadership that a variety of skills is required to carry out the myriad of tasks involved.

## 2. Materials and Methods

### 2.1. Research Methodology

An interpretive qualitative research design was adopted for this study. This comprised of semi-structured interviews to explore how principals perceived their inclusive leadership. This approach was employed as the researcher sought to understand the experiences of principals in primary schools with the purpose of deepening understanding of leadership in this context. Ethical approval was granted for the study through the host institution for this research, University College Cork (Log Number: 2021-103).

### 2.2. Data Collection Development

Interviews were chosen as the method of data collection as they would allow the researcher to explore in more depth how participants conceptualized and experienced leading autism classes. To address the main research aim [42] and ensure data collection procedures were rigorous in the research [43] a number of interview researchers were consulted in the development of a semi structured interview schedule [44,45]. An appropriate method for researching perceptions, opinions and values is the semi-structured interview [46]. The purpose of this research was to gather information of participants' perceptions and experiences as well as exploring their values which are at the core of inclusionary leadership.

*2.3. Participants*

A total of fifteen participants, ten females and five males participated in the interviews. Participants represented a mix of schools including single-sex and mixed-sex student cohorts; Catholic and non-denominational; DEIS (Delivering Equality of Opportunity in Schools, for schools whose socio-economic context would be deemed disadvantaged) and non-DEIS. Participants were recruited using calls for participation through national organizations with principal memberships.

*2.4. Data Collection*

As data were collected during the period of COVID-19 restraints, the semi-structured interviews were conducted online using Microsoft Teams, at a time chosen by the interviewee. Interviews lasted approximately 60 min. Participants were asked nine questions (see Appendix A). Audio recordings from interviews were stored securely and anonymized transcripts were created.

*2.5. Data Analysis*

Thematic analysis was the method of analysis chosen for this research [47]. Constant reflexivity and revisiting of the data allowed for patterns to emerge and developed coding. While coding allowed for a variety of initial concepts to be explored, not all concepts became categories [48]. The development of the thematic analysis procedure, where the major emerging themes were evident in several of the interviews, as well as current literature, gave validity to their interpretation and protected against researcher bias as much as possible [47].

**3. Results**

This visual representation below (Figure 1) aims to highlight the data analysis while enhancing the reading and comprehension of the study [49]. The findings are discussed thereafter.

**Figure 1.** Research findings.

### 3.1. Principal Beliefs and Values

The school leaders articulated their perspectives on how they envisioned special education within their leadership domain. Their individual beliefs and values formed the foundation of their approach. A prevalent theme among the principals was the emphasis on resources and their equitable distribution. Many expressed the view that resources should be allocated fairly, with considerations such as timetabling support to ensure equal assistance for all students. It was acknowledged that the development of the pupils in the autism classes may take a different trajectory to that of neurotypical pupils and an understanding of this was important "*but that's not to say that their development goals as such, are, aren't equal, especially for what they can achieve*" (BD). For other principals, it was important that the children who attended the autism classes were involved in all aspects of school life, with shared experiences as much as the mainstream cohort: "*that they would feel part of the group, you know, when they go to the hall for music generation, they absorb it too. . . that everything is just, you know, . . . equal*" (MH). Principals acknowledged that this vision for equity and equality was a broad aspiration.

BD spoke about opening the autism classes as a moral action as there was a need in the community for the service: "*we opened our autism classes was because there was children that should have been coming to our school. . . their siblings were going through the school. And there wouldn't have been a place in their local community for them to attend school*". BD explained that the school made the decision to "*support those families*" by opening autism classes. He went on to say that in their school "*it wasn't a case of somebody seeing or approaching us as to whether we were happy to open a class, we approached them and said, Look, we felt morally obliged, to some degree*" (BD).

### 3.2. A Broader Vision of Inclusion

The vision of inclusion and the perspectives of the principals were expressed in a variety of ways. Their ideas of inclusion were broader in terms of inclusion in the wider school context and not just the autism classes. The idea that acceptance as the core of inclusion was put forward by a number of principals: "*it's just part of our ethos. . . just one of our pillars that we will do our best to include everybody*" (FC) and "*education for all*" (MH). "*I think of inclusion, as pupils, learning together with their peers*" (KR). AW thought of inclusion as a "*kind of acceptance*" for "*children who are aware that there are children with autism or complex needs or children from different backgrounds*". SC spoke about inclusion as "*taking people's individual differences*" and she aspired to "*making it someway seamless accepting people's differences*". SC talked about her role in "*making accommodation for people so that they can. . .be the same as everybody else*". SF spoke about inclusion in school in terms of learning and she felt that every child should have "*access to the curriculum in school and appropriate access to the curriculum*". LD also put an emphasis on curriculum access for developing a model of inclusion: "*. . .so we include everybody, and we try to come up with a solution, be it a differentiated learning plan, . . ..so that everybody can access education and people of all levels of need, can access education, to the best of their ability when they're here.*". CK argued that the Irish education system pays "*lip service*" to inclusion and she believed there was a narrow vision of inclusion as "*differentiating academic work*". It is her belief that a more "*holistic approach*" (CK) needs to be taken to create inclusion. In alignment with some of the other principals, SF spoke about a wider diversity of children than just children with additional needs when outlining her vision for inclusion. She felt that "*the special needs children, the children from foreign countries, our own children, gifted children, you know, that, that everybody has access to the curriculum, and that we're providing good teaching for them*" (SF) in order for inclusion to be present. The importance of having an understanding of autism in order for inclusion to work was highlighted by DC when she stated "*first of all, inclusion will only work if people are educated. . . people need to know about autism. . .they need to understand it. . .. we've come a long way, but there's still a lot of ignorance around autism.*" CK's definition of inclusion was transformed because of the autism classes and their influence on her knowledge. She spoke about it as follows:

"I think over time, my attitude to that word has changed entirely. Primarily, probably because of the autism classes here. . . . So I would say that we're including them in other ways other than the academic way. . . .and I can definitely see inclusion." (CK).

### 3.3. Autism Classes as a Model of Distributed Leadership

One of the key ideas emerging from interviews with principals is the idea of a shared or distributed leadership practice model as being preferable when leading and managing the autism classes. Principals referred to the idea that they were in fact taking the lead from the staff within the autism classes. When it comes to the area of the autism classes many principals felt that they were at a knowledge or experience deficit given that they may never have worked in one of these classes or have prior knowledge of the area. As principal AW described: *"there's no point in me offering support in what they're doing because half the time I wouldn't know . . .and I also obviously I take my lead from them. . ."* (AW). This acknowledgement of the staff in a leadership role within the school gives rise to the idea of multiple leaders within the organization [31]. This openness, as ascribed by AW, coupled with a vision of multiple leaders, lends itself to a distributed model of leadership within the autism class.

Distributed leadership is lauded as the most beneficial style of leadership for any organization but particularly schools [50]. The vulnerability of principals when reflecting on their place within the realm of autism classes perhaps allows for a true example of distributed leadership in action in schools. Principals spoke about how the autism classes were akin to a school in its own right: *"it feels like it's almost like its own little school"* (JL), where staff had to lead and manage. The image mirrors the *"feeling of a small school"* (JL), where everyone knows each other. This idea allows staff to take ownership and responsibility for leading the learning in the autism classes. Staff working in the classes embark on professional learning to become adept in knowledge and practice in the field. This endeavor renders them more expert than the principal of the school. Most principals interviewed acknowledged and embraced this idea. The staff lead the way in the classes. The principal in a sense relinquishes the manager role to become part of the team. Their role is revolutionized. The preconception of them as all-knowing is diminished. However, they bring a set of skills to the team. Their role may be better conceived of as coach. They create space for the team to discuss difficult issues and come up with solutions. There is no hierarchy within this example. All staff levels—teacher, SNA and principal are working together with the common goal to support these autistic children. *"I work as a team approach with room seven our autism class and then talk with the teacher and the SNAs . . .what do you feel we should be doing so it's more like leading the conversation"* (MS).

The principals' responses indicate that they see themselves as a "support" to the staff and the classes. The idea of directly leading the classes was not mentioned by any participant. One principal spoke about going into the classes to gain experience and a knowledge base. While principals referred to themselves as a support for the staff within the autism classes, no principal spoke about leading from the front within this environment. It could be proffered that the principals are abdicating responsibility to the staff in the autism classes, given their open admissions to having much less expertise in the area. The idea of distributed leadership blurs the power relationships between leaders and followers [51].

### 3.4. Strategies to Promote Inclusion

Strategies such as reverse integration were used to support inclusion in schools although the COVID-19 pandemic has affected this. For example, DC commented that *"pre COVID, we did reverse integration so the children from our mainstream went down there and they cooked and so there was all of this walking up and down the corridor the whole time and they were all involved"* (DC). Supporting children as much as possible with resourcing such as Special Needs Assistants (SNAs) was put forward in support of inclusion: *". . .for me, inclusion*

*is where every child has the right to access learning in our schools, and that the resources and supports that are needed for that child are there and available*" (MS). MW defined inclusion with a similar emphasis on resourcing for access: "*inclusion for me is ensuring that every child has the opportunity and the support to access education, and any activity that's going on to the best of their ability*". LD spoke about helping children to feel included through resources such as "*SNAs in the classroom environment*" but cautioned that it is "*dependent on support, and it's dependent on having the right level of support available at a time that you really need it*". The idea of a sense of belonging was put forward as a vision for inclusion. BH spoke about "*trying to create a sense of belonging*" in the school so "*that everybody feels part of . . . the school. . . and that not only do they feel part of it, but they benefit from it*". JL talked about building a sense of belonging by trying to make "*sure that everyone in the school feels like they're a legitimate part of the school*". She spoke about organizing the school year events and that the autism classes were "*a central part of our thought process, and all decisions that we make*" (JL). BD spoke about providing a place and a welcome for every child in the local community: "*all children are welcomed into their local school, in their age-appropriate classes, and they're supported to participate in all aspects of life of the school, and I think it's about celebrating the diversity of all our students*".

*3.5. Inclusion as a Means of Diversity*

SF discussed the idea of bringing the autism classes to the school in order to promote inclusion through introducing the diversity of additional needs to the school, which was very homogenous in nature prior to this point. She explained that the school she leads is a "*very middle-class school*" and they "*would have very academic children. . . very little international children, with very little children of different faiths, and very little special needs children, . . . we do have them but we don't have the ordinary curve*". SF believed in developing inclusion though the introduction of the autism classes "*so . . . we brought these children and brought them into our school*". SF believed "*it was good for the other children as well. Because they saw well not everybody is like everyone else in the classroom*".

*3.6. Principals Envisioning of Inclusion*

The principals outlined their thoughts on how they envisioned special education in the school they were leading. Their own beliefs and values underpinned their approach. Principals acknowledged that their vision for equity and equality was a broad aspiration "*there is far more about trying to get everybody on to some sort of a level playing field, it's an equality I suppose as much as anything else*" (CK). CK cautioned that this aspiration could be the cause of a social dilemma in the future due to its complex nature

> "I think that's going to be a huge equality issue down the line, if there's going to be children who maybe could have done better in life in terms of maybe having held down a job or held down relationships or whatever... Because the support and therapy that they needed, wasn't there, there was only so much the school can do" (CK).

BD spoke about opening the autism classes as a moral action as there was a need in the community for the service, "we opened our autism classes was because there were children that should have been coming to our school. . . their siblings were going through the school and there wouldn't have been a place in their local community for them to attend school." BD explained that the school made the decision to "support those families" by opening autism classes. He proceeded to say that in their school "it wasn't a case of somebody seeing or approaching us as to whether we were happy to open a class, we approached them and said, Look, we felt morally obliged, to some degree" (BD).

It was clear in this research that the principals who participated were positively disposed to the autism classes despite the challenges and difficulties that they outlined. BD described the autism classes as "*such a positive in terms of our value system*". He went further and personally reflected that the autism classes have "*been the single best initiative that I've ever been involved in, in education. And I feel it really works*" (BD). BD has been an educator for over twenty years. MK found the experience of opening autism classes positive. She

described it as "*very wonderful, in that it's a brilliant thing to bring to the school. . .to serve your local children, your local community*" (MH). MH was very positive in terms of "*giving back to the community*". She took great pride in it and spoke about how lovely it is to "*be able to work with people who, you know, might have special needs at home and you feel like you're helping them, and that is brilliant*" (MH). The principals were eager to share stories of positivity that emanated from having the autism classes in their schools. CK told the story of a pupil based in the autism class who was preverbal. This pupil was integrating with the mainstream class but with no verbal communication. The mainstream class he was attending had a teacher who was very musical and played the guitar with the class. One Christmas the school held an assembly where the entire school body attended. There was a male pupil soprano in the school who performed for the school with great aplomb. At that point the pupil with autism approached CK the principal and asked "John[2] sing?". CK recalled that she had never heard the child speak and was unsure as to what to do. She looked towards the staff for support. The mainstream teacher interjected that John had been mouthing to an Elvis song in the class the previous week. So CK invited John to sing for the school. John sang "Love me tender" perfectly for the whole school. CK reminisced that "*every staff member was crying and half the boys as well*". She explained that the school gave the child a standing ovation and that from that moment on the child became very engaged with school. She spoke about "*how rewarding it can be to just to have a lightbulb moment*" (CK). She talked about how she wished that people had "*more stories like that*" (CK).

Kinsella [16] argues that "communication, consultation, collaboration, co-ordination and collaborative enquiry are central to the endeavor of inclusion" (p. 1354). He speaks about the processes that are essential for inclusion but this does not take into account the people involved. The principals were clear on their educational purpose [52] in relation to the autism classes and they believed that they were serving the children in their community. Many of the principals spoke about the classes in terms of their providing inclusive and meaningful education for all pupils in their local community as seen in previous research [53]. If we envisage the local school as a community of practice [54] the principals play a part alongside teachers, parents and children. All of the principals highlighted the opportunities provided to them by the autism classes. One principal described opening the autism class as the most important initiative he had ever been involved in. The principals were proud of this work in spite of the many challenges they faced. They found joy in working with children from the autism classes and were honored to share in some of their "*light bulb*" (CK) moments.

## 4. Discussion

For principals leading autism classes, the role was seen as positive and they viewed the many opportunities it afforded them. Not one of the principals interviewed expressed any regret at either opening an autism class or taking on the role even with the many challenges that they encountered. One of the principals remarked that for him, the opening of autism classes was a pivotal point in his career. The principals viewed the opportunities afforded to them by the autism classes which is indicative of their dispositions towards inclusion [55]. Their commitment to school wide inclusion was evident with the inclusive leadership styles they employed which were identified within this study. Evidence of this commitment was even more pronounced during the COVID 19 pandemic reflective of the research [56,57]. The principals reflected on the worry they felt for the pupils, their families and staff which can be found in the research [58]. The principals had a strong commitment to supporting autistic pupils by leading the classes, ensuring the correct staff were employed, sourcing resources needed and advocating for them. These stresses included encounters with parents, staffing issues and difficulties in accessing necessary resources. However, the autism classes were a great sense of pride and joy for the principals and offered them many opportunities. Some of the principals in administrative roles with non-teaching duties enjoyed the occasional opportunity to teach in the class when covering teacher release days. Being part of the community of practice in the autism

classes gave some of the principals a sense of reinvigoration. They were presented with an opportunity to upskill and learn in their own schools. This was a source of stimulation in their professional lives.

The story of John finding his voice illustrates the transformative effect the inclusion of autistic children can have on an entire school. For one principal, the classes were a way of bringing diversity to the school which had quite a homogenous pupil cohort until the special classes were formed. At no point in the interviews did any principal question the existence of the autism classes. However, one principal described how he was looking to progress inclusion even further by developing "*quasi classes*" where the children may be based but would move back and forth from mainstream as needed during the school day in a more informal basis than is currently in place. This vision for inclusion is something he is striving for within his school. There is an indication here of a move away from the traditional idea of autism classes without their extinction.

We cannot say for definite if it is the autism classes that influenced the principals to become inclusionary principals or that these principals were inclusionary prior to leading the classes and taking on this role was a natural progression. What we can say from this study is that the principals in this research who are leading autism classes are employing an inclusionary principal approach. Further to the SISL study, more research is required to dig deeper into the area of inclusionary leadership, to develop a definition to support leaders in their work. Appropriate leadership styles for inclusion have been identified but more work is needed to look at characteristics of inclusive leaders [14]. This study argues that disposition plays an important part in the inclusionary principal role. All of the principals were positively disposed to inclusion in the autism classes and most spoke about it in terms of the wider school and community sphere. The current review of the Education for Persons with Special Educational Needs (EPSEN) Act (November 2023) is timely given it is asking the opinions of staff, parents and pupils on how they view inclusion and what they would like to see in the future.

### 4.1. The Inclusionary Principal

In this study, our findings suggest that inclusionary leaders tend to employ a mix of transformational, distributed, and instructional leadership styles. They take on a variety of roles in their work, such as advocate, partner, interpreter, negotiator, helper, support, organizer and conflict resolver. They utilize these roles as and when they are needed within their work. Their leadership is underpinned by their moral values [59]. They have a strong sense of fairness and equity. They enact a servant leadership style overall. They want to serve the children, the staff, the parents and the wider community as best they can. They understand that they work within a system and use the resources they have, while striving to better the resources at their disposal. They are vulnerable in their approach. They become part of the community of practice to support everyone and develop their knowledge and skills. They illustrate a positive disposition towards including everyone in the school community. They show understanding and benevolence towards children, parents and staff. They build a culture of trust and vulnerability so everyone can develop their potential. They want to extend their knowledge through professional generosity by supporting newly appointed principals and taking part in studies such as this one. Potential for future research

This study offers empirical insights into the reality of inclusionary leadership practices in primary schools. Consistent with the literature, it indicates that principals are carrying out their role in a way that best supports students and the schools in light of their lived reality in an increasingly complex world of education and the accelerated change. The evidence highlights that it is essential that we explore new ways of thinking about leadership in order to enhance the profession and the experience for all. The current study has brought forward some issues which require further research.

While there is some leadership research from the Irish context, further research into leadership and inclusion is essential. While this study concentrates on autism classes,

further research could look into special classes in general and the broader school setting. The ideal of access for all is fast becoming a major issue in Irish education. Research into good leadership practice is needed to support this.

Theories of leadership are explored in this study. A new leadership style, inclusionary leadership, is proffered. This is built upon previous leadership models. Distributed leadership is advanced as an umbrella term rather than a singular style. Further research is needed to build on current theories of leadership and develop them so they align with the current needs of the school system.

### 4.2. Limitations to the Study

This study forms part of a doctoral thesis study, and therefore there are a number of limitations. Due to the part-time nature of the study, there were limitations as to what the researcher could achieve within the research. The number of participants (n = 15), while offering an insight, may not be representative of the full cohort of principals leading autism classes at present. The principals who did volunteer were eager to do so in order to expand the current knowledge that surrounds leading autism classes. All of the principals, while measured in their responses, were positively disposed towards the autism classes and their place in inclusion. In the future, it would be interesting to research those who are not as positively disposed to autism classes to understand a contrasting point of view.

## 5. Conclusions

This research contributes to the existing body of research of educational leadership in an Irish and International context. Traditional theoretical leadership concepts are somewhat challenged within this research with a view to the development of new ideas for inclusionary leadership. Inclusionary leadership is a new term coined from this study. It builds on prior knowledge of inclusive leadership practices [14] and traditional theories of leadership [60,61]. It signals an evolution in school leadership which parallels the changing landscape of Irish education [8,62]. There is thought and intentionality behind what leaders do when striving for inclusion in their schools [63]. The study disputes previous ideas of the hero principal [61] and proffers that principals see themselves as part of the larger school ecosystem. They understand that they have a role and the requisite skills but the top-down leadership approach is not how they carry out their work. They see value in identifying and articulating a vision to "create shared meaning and strengthen school culture through the development of collaborative practices and processes" [64] (p. 129). The inclusionary principal values working with others and in turn developing the strengths of the entire staff. To achieve this, principals recognize the central role staff play within the organization. Principals valuing and trusting their teachers in a symbiotic relationship was evident in this study [65].

Viewing the autism classes as a way to build a professional learning community [66] may provide a powerful opportunity for enhancement in education [67]. Professional learning communities are a way to build human capital within the learning organization [68]. Fitzgerald and Radford [10] (p. 995) concur in the sense that "teamwork, flexibility, collaborative and reflective practices are required to create effective learning organizations". A team approach in the autism classes is important to "reduce the sense of isolation and disperse decision making responsibility" [10] (p. 1003). The principals openly declared their inexperience in the area of autism classes but emphasized their belief in their importance. Finlay, Kinsella [69] argue that it is worrying that principals do not have the knowledge needed to support the autism classes. This study argues that while they do not feel comfortable leading in the area, they want to be part of the process of learning together with the staff. The principals self-declare as not being the most powerful person in the autism class community though they are aware that the position of principal is most powerful in the school. This separation of the person from the role of principal affords an opportunity for the power hierarchy to be broken down. All of the constituent voices of the school become more equal and therefore the principals voice speaks with

colleagues, rather than to colleagues. The principal becomes a genuine part of the team, and not the team leader. This not only is a model of distributed leadership but an example of legitimate participation where non-principal staff members get to use their voice and develop their professionalism [54]. It is contended "that teachers have the agency to lead change and to guide organizational development and improvement" [67] (p. 322). The autism classes have the potential for teachers and SNAs to guide development and improvement as the principal wholly gives over the leadership capacity in this area and enacts a "collaborative affirmation" stance [70] (p. 25). Raftery and Brennan [66] recommend the use of professional learning communities to support pupils with additional needs in Irish primary schools.

This research provides useful insights that can be transferred. The act of being an inclusionary leader is what so many principals who lead educational institutions embody. The authors argue that being an inclusionary leader should and could be transferred to any leader in any organization education or otherwise. This leadership stance would create a greater richness in any organization where people learn and work. The research has the potential to enable leaders to be the best they can be and support them as they embark on their inclusionary leadership journey.

**Author Contributions:** Conceptualization, L.D., K.C. and J.A.M.; methodology, L.D., K.C. and J.A.M.; formal analysis, L.D., K.C. and J.A.M.; writing—original draft preparation, L.D., K.C. and J.A.M.; writing—review and editing, L.D., K.C. and J.A.M.; supervision, K.C. and J.A.M. All authors have read and agreed to the published version of the manuscript.

**Funding:** This research received no external funding.

**Institutional Review Board Statement:** Ethical approval was granted for the study through the host institution for this research, University College Cork (Log Number: 2021-103).

**Informed Consent Statement:** Informed consent was obtained from all subjects involved in the study.

**Data Availability Statement:** The data presented in this study are available upon request from the corresponding author.

**Conflicts of Interest:** The authors declare no conflicts of interest. The author reports there are no competing interests to declare.

## Appendix A. Interview Questions

1. Tell me about your principal experience to date?
2. What do you understand by the term inclusion?
3. How do you perceive your role in leading and managing autism classes?
4. What are the challenges for you as the leader?
5. What are the opportunities for you as the leader?
6. Tell me about the skills or roles you utilise or take on when leading autism classes?
7. What supports are in place for you? How can you be better supported to become a more inclusive leader?
8. Is there anything you wish you knew before taking on the role? And what advice would you give to someone about to take on the role of leader of a primary school with autism classes?
9. How has the COVID 19 global pandemic impacted on your leadership particularly in relation to leadership of autism special classes?

Is there anything you would like to add or something you wish I had asked you?

## Notes

[1] Special Educational Needs Co-Ordinator.
[2] Name changed to protect identity.

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
