# Peer review of "Inclusionary Leadership-Perspectives, Experiences and Perceptions of Principals Leading Autism Classes in Irish Primary Schools"

_societies, doi:10.3390/soc14010004_

Round 1

Reviewer 1 Report

Comments and Suggestions for Authors

Even if the topic is interesting, I have some concerns regarding the methodology. There is some confusion regarding the variables that were studied and what results ultimately emerged. As the reader proceeds with the article discovers variables along the way, and even finds out about some of them when it gets to the results. Which are the research questions and what is the main purpose of this article?

It is difficult for the reader to understand which exactly were the variables studied and what results they revealed. For example, already from the abstract (line 6-7) “This research explores the perspectives and experiences of principals currently leading autism classes” (is this the purpose of this research?), and in lines 11-12 “Core beliefs and common inclusive leadership styles emerged from the interviews. The leadership styles utilised for inclusion are explored”. So, what did this research explore? Perspectives, experiences, core beliefs, leadership styles, all of them?

This manuscript would benefit if it clearly stated the purpose of the research and the sub-objectives, as well as if it maintained a consistency in the stated variables it examined.

Certainly, there is the need for proper formatting according to the journal’s instructions, however this can be done later. The references should be cited in the journal’s suggested way both in text and in the end.

Also, in line 195 it states that values are also studied, and then reaching line 300 a new variable appears which is attitudes.

The abstract is it is unstructured and unable to present to the point the essence of this study. It does not have a clear justification and there is a lack of information. Should be rewritten following the provided guidelines of the journal in a more coherent and structured manner.

Line 19, instead of the word “faith” better use “religion”, and instead of “children with additional needs” use “students with s.e.n.”, or even better disabled students (as this includes both physical and mental disabilities).

Line 24-25, regrading the definition, it worth mentioning the widely accepted definition of inclusion (General Comment 4 adopted by CRPD-UN, 2016§6) on which inclusive leadership bases its definition.

Line 55, choose another word for “anomaly”.

Furthermore, there is no need to provide such a broad definition of the different leadership styles, and instead provide more results. Also, the section of discussion should be written in a more coherent and comprehensive manner by matching the variables ultimately studied in this research

Line 231-233, this statement highlights the importance of this study. It should be transferred either in the begging or at the end, and add a little more on the importance of this research and its contribution/practical implications. Suggest authors to develop the concluding remarks. Are the concluding remarks consistent with the arguments presented? What does this manuscript add to the international scientific area compared with other published material?

Moreover, further statistical analyses (e.g., triangulation) would significantly strengthen the insights revealed by the research data or/and possible interpretation which at the moment the manuscript lacks.

Finally, there are is no clear mention of limitations and further research suggestions. It would be an advantage if certain references that date before 2012 would be replaced, if possible, by more recent studies.

Best regards,

Reviewer

Author Response

1. Summary

2. Questions for General Evaluation

Reviewer’s Evaluation

Response and Revisions

s the content succinctly described and contextualized with respect to previous and present theoretical background and empirical research (if applicable) on the topic?

Can be improved

Work completed. See comments

Are all the cited references relevant to the research?

Can be improved

Work completed. See comments

Are the research design, questions, hypotheses and methods clearly stated?

Must be improved

Work completed. See comments

Are the arguments and discussion of findings coherent, balanced and compelling?

Must be improved

Work completed. See comments

Are the results clearly presented?

Is the article adequately referenced?

Must be improved

Must be improved

Work completed. See comments

Are the conclusions thoroughly supported by the results presented in the article or referenced in secondary literature?

Must be improved

Work completed. See comments

3. Point-by-point response to Comments and Suggestions for Authors

Comments 1: Even if the topic is interesting, I have some concerns regarding the methodology. There is some confusion regarding the variables that were studied and what results ultimately emerged. As the reader proceeds with the article discovers variables along the way, and even finds out about some of them when it gets to the results. Which are the research questions and what is the main purpose of this article?

Response 1: Thank you for pointing this out. We have updated the abstract which outlines the study methodology as qualitative and the method as interview with 15 primary school principals.

Comments 2: It is difficult for the reader to understand which exactly were the variables studied and what results they revealed. For example, already from the abstract (line 6-7) “This research explores the perspectives and experiences of principals currently leading autism classes” (is this the purpose of this research?), and in lines 11-12 “Core beliefs and common inclusive leadership styles emerged from the interviews. The leadership styles utilised for inclusion are explored”. So, what did this research explore? Perspectives, experiences, core beliefs, leadership styles, all of them?

Response 2: Agree. We have, accordingly, revised the abstract to make it clear that the purpose of the research is This study explores the experiences, practices and perceptions of primary school principals currently leading autism classes in Ireland.

Comment 3: This manuscript would benefit if it clearly stated the purpose of the research and the sub-objectives, as well as if it maintained a consistency in the stated variables it examined.

Response 3: Thank you. The variables are outlined in the abstract as 15 principals.

Comment 4: Certainly, there is the need for proper formatting according to the journal’s instructions, however this can be done later. The references should be cited in the journal’s suggested way both in text and in the end.

Response 4: Thank you. Formatting is now complete to align with journal formatting.

Comment 5: Also, in line 195 it states that values are also studied, and then reaching line 300 a new variable appears which is attitudes.

Response 5: Thank you. Values and attitudes emerged from the data following analysis. They were not intentionally studied as a variable.

Comment 6: The abstract is it is unstructured and unable to present to the point the essence of this study. It does not have a clear justification and there is a lack of information. Should be rewritten following the provided guidelines of the journal in a more coherent and structured manner.

Response 6: Thank you. Abstract has been updated in line with your comments.

Comment 7: Line 19, instead of the word “faith” better use “religion”, and instead of “children with additional needs” use “students with s.e.n.”, or even better disabled students (as this includes both physical and mental disabilities).

Response 7: Thank you.  ‘Faiths’ is the term used in Ireland. Children with additional needs is the most current term used in Ireland. Both terms remain in the article.

Comment 8: Line 24-25, regarding the definition, it worth mentioning the widely accepted definition of inclusion (General Comment 4 adopted by CRPD-UN, 2016§6) on which inclusive leadership bases its definition.

Response 8: Thank you. This advice enhances the article. A quote has been included as suggested.

Comment 9: Line 55, choose another word for “anomaly”.

Response 9: Thank you. Word changed to ‘discrepancy’.

Comment 10: Furthermore, there is no need to provide such a broad definition of the different leadership styles, and instead provide more results. Also, the section of discussion should be written in a more coherent and comprehensive manner by matching the variables ultimately studied in this research

Response 10: Thank you. Included are more results which include the voice of the principals who were studied as part of the research. This supports the discussion. Discussion now matches results variables.

Comment 11: Line 231-233, this statement highlights the importance of this study. It should be transferred either in the begging or at the end, and add a little more on the importance of this research and its contribution/practical implications. Suggest authors to develop the concluding remarks. Are the concluding remarks consistent with the arguments presented? What does this manuscript add to the international scientific area compared with other published material?

Response 11: Thank you. The line highlighted has been moved to the end of the paper. Concluding remarks have been developed in line with comment 11.

Comment 12: Moreover, further statistical analyses (e.g., triangulation) would significantly strengthen the insights revealed by the research data or/and possible interpretation which at the moment the manuscript lacks.

Response 12: Thank you. Results with data from participants has now been included in results.

Comment 13: Finally, there are is no clear mention of limitations and further research suggestions. It would be an advantage if certain references that date before 2012 would be replaced, if possible, by more recent studies.

Response 13: Thank you. Limitations now included. Reference list reviewed. Current references deemed necessary for the research.

4. Response to Comments

Thank you for reviewing this paper and providing detailed comments that helped to enhance it.

Kindest Regards,

The authors

Reviewer 2 Report

Comments and Suggestions for Authors

Article Review

October 26th, 2023

Relevance / Importance of Topic

The research paper titled “Inclusionary Leadership – perspectives and Experiences of Principals leading autism classes in Irish Primary Schools" explores the core of a central educational paradigm in today's diverse society. Autism spectrum disorder (ASD) is a dominant condition, and inclusive education is vital to ensuring that every child, regardless of their abilities, can succeed within mainstream classrooms. This study offers thoughtful insights into inclusive education, specifically focusing on the roles and experiences of principals leading autism classes in Irish primary schools.

The relevance and importance of this research cannot be exaggerated. As the awareness of autism grows, understanding of perspectives and experiences becomes a foundation for shaping effective policies and practices that accommodate the diverse needs of students with autism. Subsequently, research carries importance to the academic community and not only.

Quality of analysis & evidence

Paper Strength

  1. Introduction – The author (s) presented the introduction well. This research explores the perspectives and experiences of principals currently leading autism classes. A qualitative research design was adopted using semi-structured interviews with primary school principals. The author(s) applied and highlighted several subchapters such as Leadership for inclusion in an Irish context, Language of inclusive Leadership, defining a Model of inclusive school leadership, transformational leadership, distributed leadership, instructional leadership, and combining models of Leadership. These provided comprehensive guidelines and directions for the research. This was a very detailed approach for literature introduction.

  1. Research Methodology – The author (s) used an interpretive qualitative research design for this study. This comprised semi-structured interviews to explore how principals perceived their inclusive leadership.  The author (s) had a sample for interviews such as a total of fifteen participants, ten females and five males participated in the interviews. Participants represented a mix of schools including single-sex and mixed-sex student cohorts; Catholic and non-denominational; DEIS (Delivering Equality of Opportunity in Schools, for schools whose socio-economic context would be deemed disadvantaged) and non-DEIS. Added Valur from DEI perspective.

The Authors made a practical choice in conducting quantitative vs. qualitative research in regard to the “Inclusionary Leadership – perspectives and Experiences of Principals leading autism classes in Irish Primary Schools". An interview perspective would provide more emotions but at the same time not prone to bias.

  1. Literature review – The authors used around 70 different resources which are used throughout the paper.
    1. Suggestion – author(s) are advised to present a good outline with tables to show what patterns were identified observing the resources and indicate the knowledge gap in relation to the research problem.
  2. Materials and Methods – The following key outlines from the methods used:

·       An interpretive qualitative research design with semi-structured interviews to explore how principals perceived their inclusive leadership.

·       Interviews were chosen as the method of data collection as they would allow the researcher to explore in more depth how participants conceptualized and experienced leading autism classes

·       A total of fifteen participants, ten females, and five males participated in the interviews with DEI in mind.

·       As data were collected during Covid. The semi-structured interviews were conducted online using Microsoft Teams, at a time chosen by the interviewee. Interviews lasted approximately 60 minutes. Audio recordings from interviews were stored securely and anonymized transcripts were created.

Suggestion: Author(s) are advised to indicate the number of questions that was asked, and also can we have those questions as and appending? It will add value for discussions as we do not see what questions and the impact.

  1. Discussions – discussion in the work is well presented.
    1. Suggestion – the author(s) are advised show show a mind map or table on what patterns were discovered from the interview, and how it is similar or different to literature findings.

  1. Conclusion – The conclusion sounds weak. The last paragraph is the definition which we do not really need there. We need what was achieved by the research and if any what recommendations.
    1. Suggestions – The author(s) are advised to re-write the conclusion and show the key findings and its interpretation.

Paper Description - Organization and presentation quality

The paper “Inclusionary Leadership – perspectives and Experiences of Principals leading autism classes in Irish Primary Schools" is well organized and presented with quality. The introduction is organized well, the research and methodologies are comprehensive, and the results and findings are constructive.  

Contribution to theory or practice

The research “Inclusionary Leadership – perspectives and Experiences of Principals leading autism classes in Irish Primary Schools". contributes to the theory and practice as the research on autism is fundamental.

Overall quality

Overall, the authors have done a great job with the paper and the research. The quality of the research paper “Inclusionary Leadership – perspectives and Experiences of Principals leading autism classes in Irish Primary Schools" is acceptable for the journal with minor revisions.

Author Response

1. Summary

Thank you very much for taking the time to review this manuscript. Please find the detailed responses below and the corresponding revisions/corrections in the re-submitted files

2. Questions for General Evaluation

Reviewer’s Evaluation

Response and Revisions

Is the content succinctly described and contextualized with respect to previous and present theoretical background and empirical research (if applicable) on the topic?

Yes

Thank you

Are all the cited references relevant to the research?

Yes

Thank you

Are the research design, questions, hypotheses and methods clearly stated?

Can be improved

Work completed. See comments

Are the arguments and discussion of findings coherent, balanced and compelling?

Can be improved

Work completed. See comments.

For empirical research, are the results clearly presented?

Is the article adequately referenced?      

Must be improved

Must be improved

Work completed. See comments.

Work completed. See comments.

Are the conclusions thoroughly supported by the results presented in the article or referenced in secondary literature?

Can be improved

Work completed. See comments.

3. Point-by-point response to Comments and Suggestions for Authors

Comments 1: The research paper titled “Inclusionary Leadership – perspectives and Experiences of Principals leading autism classes in Irish Primary Schools" explores the core of a central educational paradigm in today's diverse society. Autism spectrum disorder (ASD) is a dominant condition, and inclusive education is vital to ensuring that every child, regardless of their abilities, can succeed within mainstream classrooms. This study offers thoughtful insights into inclusive education, specifically focusing on the roles and experiences of principals leading autism classes in Irish primary schools.

The relevance and importance of this research cannot be exaggerated. As the awareness of autism grows, understanding of perspectives and experiences becomes a foundation for shaping effective policies and practices that accommodate the diverse needs of students with autism. Subsequently, research carries importance to the academic community and not only.

Response 1: Thank you for highlighting the strengths and importance of the study.

Comment 2: author(s) are advised to present a good outline with tables to show what patterns were identified observing the resources and indicate the knowledge gap in relation to the research problem.

Response 2: We have, accordingly, created a table (Table 1) which outlines findings from the literature.

Comment 3: Author(s) are advised to indicate the number of questions that was asked, and also can we have those questions as and appending? It will add value for discussions as we do not see what questions and the impact.

Response 3: Thank you. The number of questions is now included under ‘data collection’. Interview questions attached as an appendix to article.

Comment 4:  the author(s) are advised show a mind map or table on what patterns were discovered from the interview, and how it is similar or different to literature findings.

Response 4: Thank you. A mind map has been created at the beginning of the findings.

Comment 5: The author(s) are advised to re-write the conclusion and show the key findings and its interpretation.

Response 6: Thank you. The conclusion has been re written.

4. Response to Comments

Thank you for your very kind comments and constructive feedback.

Kindest Regards,

The authors

Round 2

Reviewer 1 Report

Comments and Suggestions for Authors

The manuscript has been sufficiently improved. I have no further observations.